# Diagnosis by Microbial Culture, Breath Tests and Urinary Excretion Tests, and Treatments of Small Intestinal Bacterial Overgrowth

**DOI:** 10.3390/antibiotics12020263

**Published:** 2023-01-28

**Authors:** Yorinobu Maeda, Teruo Murakami

**Affiliations:** 1Laboratory of Drug Information Analytics, Faculty of Pharmacy & Pharmaceutical Sciences, Fukuyama University, Sanzou, Gakuen-cho, Fukuyama 729-0292, Hiroshima, Japan; 2Faculty of Pharmaceutical Sciences, Hiroshima International University, 5-1-1 Hiro-koshingai, Kure 737-0112, Hiroshima, Japan

**Keywords:** small intestinal bacterial overgrowth, diagnosis of SIBO, conjugated bile acids, ursodeoxycholic acid, para-aminobenzoic acid, 5-aminosalicylic acid, treatments of SIBO

## Abstract

Small intestinal bacterial overgrowth (SIBO) is characterized as the increase in the number and/or alteration in the type of bacteria in the upper gastrointestinal tract and accompanies various bowel symptoms such as abdominal pain, bloating, gases, diarrhea, and so on. Clinically, SIBO is diagnosed by microbial culture in duodenum/jejunum fluid aspirates and/or the breath tests (BT) of hydrogen/methane gases after ingestion of carbohydrates such as glucose. The cultural analysis of aspirates is regarded as the golden standard for the diagnosis of SIBO; however, this is invasive and is not without risk to the patients. BT is an inexpensive and safe diagnostic test but lacks diagnostic sensitivity and specificity depending on the disease states of patients. Additionally, the urinary excretion tests are used for the SIBO diagnosis using chemically synthesized bile acid conjugates such as cholic acid (CA) conjugated with para-aminobenzoic acid (PABA-CA), ursodeoxycholic acid (UDCA) conjugated with PABA (PABA-UDCA) or conjugated with 5-aminosalicylic acid (5-ASA-UDCA). These conjugates are split by bacterial bile acid (cholylglycine) hydrolase. In the tests, the time courses of the urinary excretion rates of PABA or 5-ASA, including their metabolites, are determined as the measure of hydrolytic activity of intestinal bacteria. Although the number of clinical trials with this urinary excretion tests is small, results demonstrated the usefulness of bile acid conjugates as SIBO diagnostic substrates. PABA-UDCA disulfate, a single-pass type unabsorbable compound without the hydrolysis of conjugates, was likely to offer a simple and rapid method for the evaluation of SIBO without the use of radioisotopes or expensive special apparatus. Treatments of SIBO with antibiotics, probiotics, therapeutic diets, herbal medicines, and/or fecal microbiota transplantation are also reviewed.

## 1. Introduction

A wide variety of aerobic and anaerobic bacteria live in the human body, and the human intestine contains 100 trillion (100 × 10^12^) viable bacteria, most of which are anaerobic. These bacteria, which make up to 30% of the fecal mass, are known as intestinal flora [1]. The oral cavity contains more than 300 bacterial species, and the lower intestine contains about 500 bacterial species. The concentrations of bacteria are 10^10^ to 10^11^ colony-forming units (CFU)/mL on the tooth surfaces, 10^8^ to 10^9^ CFU/mL in saliva, and 10^11^ to 10^12^ CFU/mL in gingival scrapings, respectively [2]. The numbers of bacteria in the upper small intestine and stomach were reported to be <10^5^ CFU/mL and <10^3^ CFU/mL, respectively, in normal people, although the number in the stomach can be increased to >10^4^ CFU/mL depending on gastric pH [3]. Separately, bacterial numbers in the stomach and proximal intestine were reported to have 10^2^–10^5^ CFU/mL intestinal content [2]. Another research group reported that bacterial numbers in the small intestine (duodenum and jejunum) and in the distal small intestine (ileum and colon) were 10^3^ to 10^4^ CFU/mL and around 10^7^–10^8^ CFU/mL of the intestinal contents, respectively [4].

The number and/or type of intestinal bacteria in the intestine are modulated by various factors in favor of harmful bacteria, and clinically, the increase in the number and/or alteration in the type of bacteria in the upper gastrointestinal tract is regarded as small intestinal bacterial overgrowth (SIBO). SIBO is induced by malabsorption (or maldigestion) of foodstuffs under abnormal intestinal anatomy, impaired gastric acid secretion, and/or impaired intestinal motor function as follows: surgical operations of the stomach or small intestine, liver or kidney diseases, pancreatic insufficiency, pernicious anemia, inflammatory bowel diseases (IBD) such as Crohn’s disease, cancer, cirrhosis, antibiotic therapies, immune disorders, emotional stress, poor diet, radiation, ageing, sex (female > male), medical use of proton pump inhibitors (PPIs) or opioids, and so on. SIBO accompanies various bowel symptoms such as abdominal pain, bloating, gases, diarrhea, and so on. [1,5,6,7,8,9,10,11,12].

This article reviews diagnostic methods of SIBO by focusing on the use of bile acid conjugates as diagnostic substrates, in addition to the quantitative cultural analysis of duodenum/jejunum fluid aspirates and breath tests (BT). The review was made mainly from the viewpoint of the pharmacokinetics of diagnostic substrates and the modulation by intestinal bacteria (SIBO) as an influencing factor. BT detects the increase in the concentration/rate of hydrogen (H_2_), methane (CH_4_) and/or carbon dioxide (CO_2_) gases in the breath after ingestion of carbohydrates or lipids, as compared with control (pre-ingestion). Bile acids have a close and important association with small intestinal bacteria with respect to the components of bile acids (metabolism), enterohepatic circulation, and hosts’ health/disease states [8,13,14,15,16,17,18]. The literature was searched using PubMed, Google Scholar, Online search engines and appropriate keywords such as SIBO, diagnosis and treatments. Many articles are available, including review articles, clinical consensus, and guidelines, for the bacterial culture of duodenum/jejunum fluid aspirates and BT for SIBO diagnosis [7,10,19,20,21,22]. In contrast, the review article regarding the urinary excretion tests of bile acid conjugates for the SIBO diagnosis is not yet available.

## 2. Interaction of Endogenous/Exogenous Compounds with Intestinal Bacteria

Intestinal bacteria contain various enzymes, such as β-glucosidase, β-glucuronidase, arylsulfatase, bile salt (or cholylglycine) hydrolase, nitroreductase, and nitrate-reductase. Intestinal bacteria also provide a suite of additional reactions, including acetylation, deacylation, decarboxylation, dihydroxylation, demethylation, and dehalogenation [23,24]. In general, oral bioavailability (F) is expressed as a product of the fraction absorbed (Fa), the fraction that avoids the intestinal first-pass metabolism (Fg), and the fraction that avoids the hepatic first-pass metabolism (Fh), with the following relationship: F = Fa × Fg × Fh. The intestinal first-pass metabolism is expressed as the intestinal extraction ratio (Eg) and the hepatic first-pass metabolism is the hepatic extraction ratio (Eh); that is, F = Fa × (1 − Eg) × (1 − Eh) [25,26]. For drugs that undergo Phase 2 conjugate metabolism and enterohepatic circulation, the hydrolytic metabolism to parent compounds by intestinal bacteria will be involved in the bioavailability. This would indicate that intestinal-bacteria-mediated metabolism can exert a marked impact oral bioavailability and pharmacological action of a few specific drugs with enterohepatic circulation [27,28,29,30,31,32]. For example, many dietary glycosides are enzymatically hydrolyzed by intestinal bacteria, and the aglycon produced is absorbed and conjugated to its glucuronide and/or sulfate in epithelial cells and the liver. The glucuronide/sulfate conjugate metabolites are excreted into the intestinal lumen via the bile duct are hydrolyzed again to their aglycon and then undergo enterohepatic circulation repeatedly [28]. In the case of non-steroidal anti-inflammatory drugs (NSAID), for example, the inhibition of bacterial β-glucuronidase-mediated cleavage of NSAID glucuronides in the small intestinal lumen can protect against NSAID-induced enteropathy caused by locally high concentrations of NSAID aglycones [27]. As such, the modulation of intestinal bacteria can modulate the pharmacokinetics, and therefore, pharmacodynamics of various exogenously ingested substances, including drugs and dietary substances (especially polyphenols), and therefore, affect the host’s health/disease states. Diagnosing SIBO is important, as treating creates a safer and more reliable pharmacotherapy and health.

The pharmacokinetics and pharmacological action of bile acids are also closely associated with small intestinal bacteria. Primary bile acids, cholic acid (CA), and chenodeoxycholic acid (CDC), are synthesized from the cholesterol, and some parts of these bile acids are conjugated either with glycine or taurine in the liver. These amino-acid-conjugated bile acids are secreted into the intestinal lumen via the bile duct and deconjugated by intestinal bacterial bile salt hydrolase (cholylglycine hydrolase, EC 3.5.1.24) to former primary bile acids. The primary bile acids are metabolized to the secondary bile acids, deoxycholic acid (DCA), lithocholic acid (LCA), and ursodeoxycholic acid (UDCA) by intestinal bacteria and these conjugated/unconjugated bile acids are absorbed again to be undergoing enterohepatic circulation [15,33]. Bile acids are solubilizers of hydrophobic compounds and potent antibacterial compounds, and play an important role in shaping the intestinal microbial composition, cell signaling, and ecology, in addition to the modification of bile acids through deconjugation, dihydroxylation, dehydrogenation, and epimerization of the cholesterol core [8,32,34,35]. Bile acid and intestinal bacteria have important and close relationships with each other as a determinant of the health/disease status of the host pharmacokinetically and pharmacologically.

## 3. Diagnosis of SIBO by Cultural Analysis of Bacteria

Cultural analysis of bacteria in duodenum/jejunum fluid aspirates is regarded as the golden standard for the diagnosis of SIBO. In this section, cultural analysis, analysis by 16S rRNA gene sequencing of bacteria in aspirates, and the analysis of bacterial 16S rRNA/rDNA gene fragments in plasma are reviewed.

### 3.1. Cultural Analysis of Bacteria in Duodenum/Jejunum Fluid Aspirates

To diagnose SIBO clinically, microbial culture in duodenum/jejunum fluid aspirates is carried out to detect bacterial numbers (colony-forming units of bacteria per mL of aspirate, CFU/mL) and/or composition of mucosal microbiota. Recently, new terms including “intestinal methanogen overgrowth” and “small intestinal fungal overgrowth” have also been introduced to emphasize the contribution of CH_4_ production by methane-producing archaea and fungi in small intestinal dysbiosis [12,36]. However, variation is observed in the criteria, or the cut-off level of CFU/mL in diagnosing SIBO by a quantitative cultural method among research groups. For example, some articles reported that the number of bacteria of >10^5^ CFU/mL of duodenum/jejunum aspirates was considered positive [37,38,39]. Another group reported that aspirate culture was considered positive if >10^5^ CFU/mL of Gram-negative flora was identified on culture, and contaminated if >10^5^ CFU/mL of Gram-positive flora was identified [38]. Additionally, it was reported that ≥10^3^ CFU/mL was considered to suggest positive, particularly if colonic-type bacteria were present in the upper intestine [21]. Recently the cut-off levels of >10^3^ or ≥10^3^ CFU/mL were reported for the definition of SIBO [40,41,42]. It is considered that survival rates of intestinal bacteria can vary possibly depending on bacteria species, cultural methods and/or the patient’s gastrointestinal condition. For example, it was reported that duodenal samples from SIBO subjects had 4 × 10^3^-fold higher counts than non-SIBO subjects when plated on MacConkey agar (*p* < 0.0001) and 3.8-fold higher counts when plated on blood agar (*p* < 0.0001) [41]. Although the diagnosis by cultural analysis for SIBO using duodenum/jejunum fluid aspirate is accepted as the gold standard of SIBO diagnosis, this method is reported to be high cost, invasive, and not without risk to the patient [5,43,44,45,46]. In cultural analysis, the aseptic technique was reported to be critical in minimizing cross-contamination, and the collecting methods of aspirates, including the lumen catheter, are reviewed in detail [45].

### 3.2. Analysis by 16S Ribosomal RNA/DNA Gene Sequencing

Analysis of bacteria in duodenum/jejunum fluid aspirates by 16S ribosomal RNA (rRNA) gene sequencing for identification, DNA-based cell counting, and characterization (function) of bacteria is also carried out for SIBO diagnosis [41,42,44,47]. When the combined bacterial colony counts ranged from 5.7 × 10^3^ to 7.9 × 10^6^ CFU/mL of jejunum aspirate obtained from patients reporting symptoms consistent with SIBO, DNA-based analysis yields ranged from 1.5 × 10^5^ to 3.1 × 10^7^ bacterial genomes per mL, and the microbial viability ranged from 0.3% to near 100%. In addition, when the bacterial composition of feces and mucosa of the duodenum and the sigmoid colon was determined by 16S rRNA-amplicon-sequencing, the bacterial profiles of feces and the mucosa of the sigmoid colon, but not of the duodenum, were different between irritable bowel syndrome patients and healthy subjects [44]. SIBO patients having a bacterial count ≥ 10^3^ CFU/mL were characterized by an increased relative abundance of *Streptococcus* and decreased relative abundance of *Bacteroides* compared with non-SIBO patients [42]. Increases in Gammaproteobacterial in SIBO patients having >10^3^ CFU/mL resulted primarily from higher relative abundances of the family *Enterobacteriaceae*, which correlated with the symptom of bloating. Increases in family *Aeromonadaceae* correlated with urgency with bowel movement [41]. Separately, relative abundances of a mucosal *Clostridium* spp. and an aspirate *Granulicatella* spp. were higher in coliform SIBO vs. non-SIBO subgroups in which SIBO patients were defined as ≥10 CFU/mL coliform or ≥10 CFU/mL upper aerodigestive tract bacteria. It was stated that culture-based results of small bowel aspirates do not correspond to aspirate microbiota composition but may be associated with the species richness of the mucosal microbiota [47]. Analysis by 16S rRNA gene sequencing of bacteria would provide detailed information on the microbiota number and composition in aspirates. In addition, the detection of bacterial 16S rRNA/16S rDNA gene fragments in plasma is also reported, in which the quantity of 16S rRNA/16S rDNA gene fragments in plasma could be a marker of systemic microbial translocation from a ‘leaky’ gut under pathogenesis of various diseases, such as HIV, type II diabetes and hepatitis, and also the intestinal bacterial number and composition. It was also reported that the bacterial 16S rDNA assay can analyze 90% of bacterial strains, including Gram-positive and Gram-negative bacteria. However, the use of this assay is highly challenging because of its high technical demands and the risk of contamination [48,49].

### 3.3. Cultural Analysis of Fecal Bacteria

The fecal bacterial composition of children with or without SIBO was analyzed, in which SIBO was screened through H_2_/CH_4_ BT with lactulose. SIBO was identified in 61.0% of the children. Children with SIBO presented a significantly higher frequency and count of *Salmonella*, and lower counts of *Firmicutes* and total *Eubacteria* when compared to those without SIBO [50]. The stool of 2-year-old children with SIBO was analyzed by 16 S V4 rDNA microbiome analysis in which glucose H_2_ BT positivity in children was suggested to be due to an upper intestinal *Lactobacillus bloom*, potentially explaining the association of SIBO with the gut damage and inflammation that leads to malnutrition [51]. To assess SIBO in IBS, the bacterial composition of feces and the mucosa of the duodenum and sigmoid colon was determined by 16S rRNA-amplicon-sequencing. The bacterial profiles of feces and the mucosa of the sigmoid colon, but not of the duodenum, differed between IBS patients and healthy subjects. The fecal bacterial profile differed between IBS subtypes, while the mucosa-associated bacterial profile was associated with IBS symptom severity and BTs results at baseline (H_2_ and/or CH_4_ ≥ 15 parts per million (ppm)). It was concluded that while BTs reflected the mucosa-associated bacterial composition, there was no evidence of a high prevalence of SIBO or small intestinal bacterial alterations in IBS [52]. The diversity and composition of fecal microbiota in patients with functional abdominal bloating and distention (FABD) and healthy individuals were compared to evaluate the relationship between SIBO and dysbiosis. The fecal microbiota profiles in FABD patients were different from those in healthy controls, particularly in SIBO-positive patients. Significantly more abundant *Prevotella copri* and *F. prausnitzii*, and significantly less abundant *B. uniformis* and *B. adolescentis* were observed in SIBO-positive patients, compared with healthy controls, suggesting a role of gut microbiota in the pathogenesis of FABD [53]. Similarly, fecal microbiome analyses demonstrated differences between systemic sclerosis patients with and without SIBO and differences in the diversity of species between healthy controls and patients with systemic sclerosis [54]. The microbial communities of mucosal specimens from the duodenum, ileum, sigmoid colon, and feces of patients with and without SIBO were analyzed, in which SIBO was diagnosed by lactulose BT. The bacterial compositions and their abundance of the mucosal samples of the duodenum, ileum, and sigmoid colon were significantly different between patients with and without SIBO. However, four genera (*Lactobacillus*, *Prevotella_1*, *Dialister*, *and norank_f_Ruminococcaceae*) had similar changes among the duodenal, ileac, and sigmoid colonic samples in the SIBO-positive subjects compared to the SIBO negative subjects. It was reported that mucosa-associated taxa that may be potential biomarkers or therapeutic targets of SIBO were identified, although further study is needed on their mechanisms and roles in SIBO [55].

## 4. Diagnosis of SIBO by Breath Tests

In diagnosing SIBO, breath tests (BT) are commonly and widely used in clinical settings. BT detects excretion rates of hydrogen (H_2_), methane (CH_4_), and/or carbon dioxide (CO_2_) gases are excreted in the breath after ingestion of a fixed amount of diagnostic substrate compounds such as carbohydrates or lipids. BT using C-labeled compounds is also used to evaluate specific (microsomal, cytosolic, and mitochondrial) hepatic metabolic pathways in liver diseases and to assess gastric emptying kinetics [56]. In SIBO patients, the higher excretion rates of H_2_/CH_4_ gases are expected due to maldigestion (or malabsorption) and increased bacteria-mediated fermentation of orally ingested substrate compounds compared to those in healthy subjects. In this section, diagnosing substrate compounds for BT was reviewed by dividing into two groups: unlabeled and isotope-labelled diagnostic substrates.

### 4.1. Use of Hydrocarbons

Different from the case of cultural analysis of duodenum/jejunum fluid aspirates, clinical consensus and guidelines are available for H_2_ and CH_4_-based BTs in many countries [7,10,19,21,22]. In the North American Consensus, for example, the doses of lactulose, glucose, fructose, and lactose for diagnosing SIBO are 10, 75, 25, and 25 g, respectively, and a rise in H_2_ of ≥20 ppm by 90 min from baseline during glucose or lactulose BT is considered positive, and a rise in CH_4_ levels ≥ 10 ppm from baseline is considered positive [10]. D-xylose, sorbitol, sucrose, and inulin are also reported as diagnostic substrates of BTs in which inulin was used mostly for the assessment of orocaecal transit time (OCTT, the time from when food is eaten till it reaches the large intestine). Significantly higher OCTT is observed in patients with SIBO and IBD or SIBO and type 2 diabetes as compared to SIBO-negative patients [9,19,57,58,59].

The results of BTs could vary depending on the gastroenterological symptoms and/or disease states. For example, it was reported that lactulose (10 g) and glucose (80 g) H_2_ BTs were only abnormal in 1 out of 17 subjects, whereas the 1 g ^14^C-D-xylose BT was abnormal in 6 out of 17 elderly hypochlorhydria subjects. Thus, non-invasive BTs for bacterial overgrowth were considered not reliable in subjects with hypochlorhydria [57,60]. In addition, the H_2_ gases produced by a carbohydrate are consumed to form CH_4_ gases by methanogenic archaea (not bacteria) during pregnancy and/or to form H_2_S gases by sulfate-reducing bacteria. Thus, it was suggested that measuring CH_4_ concentrations has an added value to the ^13^C/H_2_ BT to identify methanogenic subjects with lactose malabsorption or SIBO [7,55,61].

### 4.2. Use of Carbon Isotope-Labelled Substrates

Carbon isotope-labelled compounds such as ^14^C-, radioactive isotope, or ^13^C-, non-radioactive stable isotope, labelled D-xylose, lipids such as triolein, palmitic acid, acetate, propionate, and ureide (or N-acyl urea) were used as diagnostic substrates of BT for SIBO. Among these C-labeled compounds, lipids BTs were used mainly to evaluate the malabsorption of lipids [60,62]. The ingested isotope-labelled compounds are metabolized and exhaled ^14^CO_2_/^13^CO_2_ gases are measured by mass spectrometry or infrared spectroscopy. The stable isotope ^13^C is reported to be preferred to the radioactive isotope ^14^C, and ^13^C-BTs are valuable, non-invasive diagnoses that can be widely applied for the assessment of gastroenterological symptoms and diseases. Regarding this, it was also reported that the effective dose of ^14^C-D-xylose was estimated to be 0.07 mSv/MBq for BT. Thus, from a radiation protection point of view, no need was found for restrictions in using the ^14^C-labeled D-xylose on adults with the activities normally administered (0.07–0.4 MBq) [63]. Other research groups reported that the stable isotope technique presents an elegant, non-invasive diagnostic tool promising further options for clinical applications [64,65].

#### 4.2.1. Use of ^14^C/^13^C-D-xylose

The 1 g ^14^C-D-xylose BT, utilizing a substrate with more predominant absorption in the proximal small intestine and which can be catabolized by Gram-negative aerobic bacteria, appears to have a greater degree of sensitivity and specificity than the bile acid BT in detecting the presence of SIBO [66]. The sensitivity of the ^14^C-D-xylose BT for bacterial overgrowth was examined in patients, and it was not a suitable alternative to the culture of aspirate for the investigation of subjects for bacterial overgrowth [67]. The accuracy of ^14^C-D-xylose BT for diagnosing SIBO was evaluated in patients with severe gastrointestinal motor dysfunction (the lack of consistent delivery of ^14^C-D-xylose to the bacterial contamination region). Higher sensitivity and specificity were obtained with ^14^C-D-xylose BT by considering the delayed gastric emptying rate of ^14^C-D-xylose as a correcting factor in the detection of SIBO [66,68]. Similarly, the accuracy of ^14^C-D-xylose BT was evaluated by comparing it with the 50 g H_2_ glucose BT in patients with severe gastrointestinal motor dysfunction and suspected SIBO. There was no significant difference between the two tests, but a tendency in favor of the 50 g glucose H_2_ BT was observed [69]. The use of stable isotopes has extended the range of BT applications to include pediatric and obstetric subjects [70]. ^13^C-D-xylose BT was performed in patients with different disease histories of alcohol overconsumption, coeliac disease, or a functional bowel disorder, in addition to healthy controls, by dividing the time curves of ^13^CO_2_ excretion in breath samples into two phases, the small intestinal absorption phase (0–60 min) and the colonic microbial metabolism phase (90–240 min). The results suggested that patients with a history of alcohol overconsumption suffer from both small intestinal malabsorption and impaired colonic microbial metabolism, although further investigation was needed to clarify the role of gut microbiota in chronic alcohol overconsumption [71].

#### 4.2.2. Use of lactose-^13^C-ureide

The usefulness of the lactose-^13^C-ureide BT (^13^C-LUBT) in diagnosing SIBO was evaluated by comparing it with the glucose H_2_ BT and cultures of intestinal aspirates in patients with suspected SIBO in which patients with more than 10^6^ CFU/mL or the presence of colonic flora were defined as culture positive. The use of ^13^C-LUBT, or expiratory isotopic recovery, can assess OCTT noninvasively, and its relationship to the gastric emptying rate and small intestinal transit by avoiding the contribution of endogenous CO_2_ gases in the breath. The LUBT (2 g dose) had a sensitivity of 66.7% and a specificity of 100% to predict SIBO positives. In contrast, the sensitivity and specificity of glucose H_2_ BT (50 g dose) were 41.7% and 44.4%, respectively. Thus, the LUBT was superior to the glucose H_2_ BT for detecting SIBO [64]. In addition to the diagnostic substrate for SIBO, LUBT has mostly been used as a reliable, non-invasive substrate for the assessment of OCTT [58,59,72].

#### 4.2.3. Use of glycine-1-^14^C-labeled Glycocholate

Isotope-labelled glycocholate (glycine-1-^14^C-labeled glycocholate, ^14^C-GCA) has been used as a substrate in the clinical diagnosis of SIBO [73]. When ^14^C-GCA was administered to patients with ileal dysfunction (ileitis or ileal resection), cholesterol cholelithiasis with or without cholecystectomy, and control subjects, ^14^CO_2_ gases were excreted into the breath more rapidly in all the patients with ileal dysfunction than in the control subjects. In the patients with ileal dysfunction, 78.3% of the dosed ^14^C was excreted into the breath within 12 h, while 39.1% of the dose was excreted in the control subjects, indicating that the sensitivity of ^14^C-GCA BT for diagnosing ileal dysfunction was increased by determining the rate of the excretion of the breath ^14^CO_2_ [73]. When ^14^C-GCA BT was performed in patients clinically suspected of SIBO with functional impairment of the ileum and diarrhea, early and highest ^14^CO_2_ expiration peaks were found in patients with fistulae between the proximal small intestine and colon [74]. In patients with SIBO, the diagnostic efficacy was compared between ^14^C-GCA and ^14^C-D-xylose BTs. It was found that ^14^C-D-xylose BT appears to have a greater degree of sensitivity and specificity than ^14^C-GCA BT in detecting the presence of SIBO [66]. The performance of the ^14^C-GCA BT as an indicator of bacterial colonization of the jejunum was compared with jejunum aspirate cultural analysis. Ninety-one of the 145 cultures (62.8%) were positive, while only 31 (21.4%) of the BT were positive [75]. Simultaneous ^13^CO_2_ BT and fecal assays for the detection and quantitation of intestinal malabsorption after ingestion of ^13^C-GCA were performed in children. One child with SIBO had an abnormal BT and a normal fecal test, and both the BT and fecal test were abnormal for a child who had undergone an ileal resection [70]. Patients with liver cirrhosis were tested by ^14^C-GCA BT. Expired breath samples from patients with liver cirrhosis and SIBO showed a marked increase of ^14^CO_2_-specific activity, and administration of chloramphenicol to these patients significantly reduced ^14^CO_2_-specific activity, indicating the contribution of SIBO in the increased ^14^CO_2_ expiration. The SIBO was confirmed by the cultural analysis of aspirates [76]. After oral ingestion of ^14^C-GCA, five patients with conditions associated with stasis and bacterial overgrowth in the small intestine, and 17 of 19 patients with ileal resection and documented severe bile acid malabsorption showed strikingly higher ^14^CO_2_-specific activity as compared with healthy control subjects. Measurement of fecal ^14^C permitted complete separation of bile-acid malabsorption from SIBO. ^14^C-glycine is released from ^14^C-GCA by the intestinal bile salt hydrolase, and the released ^14^C-glycine is absorbed, metabolized in the liver, and finally, eliminated through breath as^14^CO_2_. The un-deconjugated ^14^C-GCA in the small intestine enters the large intestine, is deconjugated by colonic bacteria, and released ^14^C-glycine is excreted and measured in the feces. It was concluded that the ^14^C-GCA BT is a rapid simple tubeless outpatient procedure for detecting increased deconjugation of bile acids in intestinal disease [14,77].

## 5. Development of Bile Acid Conjugates for SIBO Diagnosis

Un-conjugated and conjugated bile acids have a tight association with the function of small intestinal bacteria pharmacokinetically and pharmacologically. This indicates that the modulation of small intestinal bacteria, such as SIBO, can alter the pharmacokinetics and pharmacological action of bile acids and the health/disease states of hosts by changing the bacterial number and composition. In developing bile acid conjugates, CA or UDCA was used as a bile acid (carrier), and PABA or 5-ASA was used as an analytical marker in evaluating intestinal bacteria activity by considering their biological and pharmacokinetic properties. In the case of 5-ASA, 5-ASA-bile acid conjugates were also used as the delivery system of 5-ASA to the large intestine since 5-ASA is a clinically important anti-inflammatory agent. In this section, the biological and pharmacokinetic properties of these bile acid conjugates and conjugating components (PABA, 5-ASA) are reviewed.

### 5.1. Biological and Pharmacokinetic Properties of Bile Acids

Urinary excretion tests of bile acid conjugates for SIBO diagnosis were developed by synthesizing PABA-CA, PABA-UDCA, PABA-UDCA disulfate and 5-ASA-UDCA monophosphate. Both CA, a trihydroxy-bile acid, and UDCA, a dihydroxy-bile acid, exhibited low solubilizing activity for Sudan III, a lipophilic compound that forms hydrophobic interactions with the hydrocarbon chains of lipids, and low rectal membrane permeability-enhancing effects in rat rectums. This would indicate that both CA and UDCA are less cytotoxic bile acids compared to other dihydroxy bile acids such as deoxycholic acid (DCA) and chenodeoxycholic acid (CDCA) [34]. Among various bile acids, UDCA, or commercially available *Ursodiol*^®^, is used to treat a variety of hepatic and gastrointestinal diseases, including cholesterol gallstone; because UDCA can solubilize cholesterol, it inhibits hepatic cholesterol synthesis and its secretion into bile, and increases the bile flow rate, as well as other bile acids [76,77,78]. In addition, UDCA and its primary metabolite, lithocholic acid (LCA) can protect against colonic inflammation in a mouse model [79,80,81], and partially restore intestinal microbiota, repair intestinal barrier integrity, and attenuate hepatic inflammation in the non-alcoholic steatohepatitis mouse model [82]. When patients with SIBO and dyspeptic symptoms were treated with UDCA (100 mg each, three times/day) for two months, the total CH_4_ gas levels for 90 min of the UDCA-treated group significantly decreased compared with baseline as diagnosed by lactulose BT [83]. UDCA treatment can reduce itch and lower endogenous serum bile acids in intrahepatic cholestasis of pregnancy and increase intestinal microbiota with Bacteroidetes and bile salt hydrolase activity [84]. In Figure 1, the chemical structures of UDCA, PABA-UDCA, PABA-UDCA disulfate, and 5-ASA-UDCA monophosphate are presented.

Oral administration of UDCA produced a unique bile acid profile with a high abundance of tauro-UDCA (TUDCA) and glycol-UDCA (GUDCA) and significantly accelerated bile acid enterohepatic circulation [85]. In addition, UDCA and TUDCA can inhibit *Clostridioides difficile* toxin-induced toxicity in vitro and in vivo [78,84,85,86]. All conjugated bile acids showed active transport in the following order: TCA > TUDCA > TCDCA in the rabbit’s intestine. Unconjugated bile acids including UDCA, except CA, are passively absorbed by the intestinal membrane, and the rates of unconjugated bile acids were in the following order: DCA > CDCA > UDCA > CA in which CA showed both passive uptake and active transport [87,88]. The intestinal absorption of UDCA is slow and incomplete, possibly due to the low solubility in the intestinal contents. When subjects bearing occluding balloons received UDCA orally at a dose of 250, 500, or 750 mg, 21–50% of the ingested doses were recovered in solid form in jejunum contents due to the poor solubility of UDCA in the gastro-duodenal-jejunal lumen of fasted subjects [89]. By increasing the dose of UDCA, the absorption decreased as follows: in patients who had a complete extrahepatic biliary obstruction caused by pancreatic carcinoma but no intestinal or liver disease, the average absorption rates after oral administration of 250 mg, 500 mg,1000 mg, and 2000 mg of UDCA were 60.3%, 47.7%, 30.7%, and 20.8%, respectively [90] Patients with pancreatic carcinoma and extrahepatic biliary drainage received 750 mg UDCA in three divided doses, and absorption of UDCA increased from 39.8% on Day 3 to 61.1% on Day 10 of the administered dose, indicating an improvement of the absorption rate after the decrease of cholestasis by 53.7% [91]. Similarly, when patients with complete extrahepatic biliary obstruction caused by pancreatic carcinoma but without intestinal or liver disease received UDCA at a dose of 750 mg, the oral bioavailability was 55.1% [92].

### 5.2. Biological and Pharmacokinetic Properties of PABA

As the diagnostic test of SIBO in laboratory animals, an enzyme-labile substrate consisting of PABA conjugated to a bile acid was synthesized, in which in the presence of enteric bacteria, PABA was expected to be split from the bile acid conjugates, absorbed rapidly, and excreted in the urine [93]. PABA is a vitamin B cofactor and is absorbed by a non-saturable, sodium-independent and passive diffusion, like other vitamin B members, and jejunal and ileal absorption rates were similar [94]. PABA fits well as a building block for a general chemical library of “drug-like” molecules with a wide range of functional and structural diversity [95]. PABA was reported as an excellent candidate for development as a bacteria-specific imaging tracer as it is rapidly accumulated by a wide range of pathogenic bacteria, including metabolically quiescent bacteria and clinical strains, but not by mammalian cells [96]. Recently, it was also reported that PABA is rapidly accumulated in a wide range of pathogenic bacteria, motivating the development of related positron emission tomography (PET) radiotracers [97].

N-benzoyl-L-tyrosyl-p-aminobenzoic acid (PT-PABA), a derivative of PABA, is used for a simple exocrine pancreatic function diagnostic (PFD) test because PT-PABA is hydrolyzed by chymotrypsin in pancreatic juice and the released PABA is absorbed. It was reported that when 500 mg BT-PABA was ingested, the recovery of PABA in the urine was significantly lower in patients with calcifying chronic pancreatitis (58.6%) or with noncalcifying chronic pancreatitis (68.6%) as compared with in healthy normal subjects (81.0%) [98]. Additionally, a significantly low urinary excretion rate of PABA in the PFD test as compared with the controls was reported in patients with primary diabetes mellitus and decreased in cases of chronic pancreatitis [99,100,101].

Regarding intestinal absorption, PABA was reported to be taken up in a linear fashion in the intestinal mucosa and its effective permeability coefficient indicated 100% absorption [99]. Additionally, the average recovery in 24 h urine collections for the PABA tablet, aminobenzoate potassium capsule, and PABA/aminobenzoate potassium in food was 98.8%, 95.1%, and 93.2%, respectively, in healthy volunteers [102]. PABA is metabolized to N-acetyl-PABA (Ac-PABA) by arylamine N-acetyltransferase (NAT) in rat and human liver and intestine, and saturation and substrate inhibition were observed at higher PABA concentrations [99]. When healthy Chinese volunteers ingested 200 mg PABA, the mean 24 h urinary recovery of PABA, PABA-COOH conjugates, Ac-PABA, and Ac-PABA-COOH conjugates were 2.9%, 5.2%, 13.9% and 42.9% of the ingested dose, respectively. These Chinese data are fairly different from those reported in one Caucasian subject from the same research group, for example, p-amino hippuric acid (PAH), the main metabolite in a Caucasian was not identified in Chinese volunteers [103,104], implying the presence of race difference in PABA metabolism.

### 5.3. Biological and Pharmacokinetic Properties of 5-ASA

5-ASA (or mesalazine) is an anti-inflammatory agent for the treatment of IBD. Both Crohn’s disease and ulcerative colitis are involved in IBD, and these diseases mostly occur in the large intestine, although Crohn’s disease can occur anywhere from the mouth to the anus. When 5-ASA alone was administered orally, 5-ASA was not effective in treating IBD because 5-ASA was rapidly absorbed in the upper part of the gastrointestinal tract. To deliver 5-ASA to the colon efficiently, various slow-release prodrug-utilizing bacteria-mediated azo reduction, such as balsalazide, salazopyrine, sulphasalazine, olsalazine and mesalazine have been developed [105,106,107]. 5-ASA is metabolized to N-acetyl-5-ASA (Ac-5ASA) and some other phase II metabolites, such as N-formyl-5-ASA, N-butyryl-5-ASA, and N-beta-d-glucopyranosyl-5-ASA [108,109].

Regarding the membrane transport mechanism, it was reported that the transport of total 5-ASA (parent drug plus intracellularly formed Ac-5ASA) was linear with time, concentration- and direction-dependent, as well as higher secretory transport was mainly caused by higher transport of the metabolite (suggesting efflux transport of metabolite). Transport of 5-ASA (only parent drug) was saturable (transepithelial carrier-mediated) at low doses, dominated by passive, paracellular processes in higher doses [110]. The intestinal epithelial membrane expresses ATP-binding cassette (ABC) efflux transporters, such as P-glycoprotein (P-gp), multi-drug resistance-associated proteins (MRPs), and breast cancer resistance protein (BCRP), in addition to various solute carrier (SLC) influx transporters. These ATP-dependent efflux transporters, especially P-gp, can decrease the oral bioavailability of various clinically available substrate compounds depending on their affinity [111,112]. 5-ASA was not a substrate of P-gp and MRP2 in Caco-2 cells, but Ac-5ASA was found to be a substrate of MRP2, and the efflux transport was significantly higher than the influx transport. The efflux transport of Ac-5ASA was significantly inhibited by quercetin, an MRP2 inhibitor, in Caco-2 cells [113,114,115]. Separately, it was also reported that 5-ASA transport was saturable (transepithelial carrier-mediated) in low doses, dominated by the passive, paracellular process in higher doses, and the involvement of sodium-coupled monocarboxylate transporter (SMCT1) in the mouse colonic mucosa [116]. In addition to the epithelial membrane of enterocytes, P-gp, MRP2, and BCRP are expressed on the bile canalicular membranes of hepatocytes. In addition, MRP3 is also expressed on the basolateral membrane of hepatocytes and transport substrates compounds from hepatocytes to the blood circulation. Thus, in the case of MRP substrate compounds, they would efflux from the hepatocytes into the bile by MRP2-mediated transport and/or into blood circulation by basolateral MRP3-mediated efflux transport depending on the affinity of compounds to each transporter. Thus, hydrophilic MRP substrate compounds would also be excreted into the urine, at least partly [111,117].

In the single oral dose regimen, systemic absorption of 5-ASA and Ac-5ASA was low and did not differ between the following three groups: healthy volunteers, patients with ulcerative colitis, and patients with Crohn’s disease in remission. Only about 20% of the 5-ASA given was absorbed, with more than 80% of the drug being available in the terminal ileum and colon for therapeutic activity [118]. When Pentasa^®^ (a controlled release 5-ASA preparation) was administered to healthy children at a dose of 1000 mg/day for 6 days, the urinary excretion of 5-ASA and Ac-5ASA after Pentasa^®^ was higher than after Salazopyrin^®^ (a prodrug of 5-ASA containing azo-bond) (32% vs. 25%) [119]. When 5-ASA was administered to the stomach, small intestine or ileocecal region in healthy volunteers, the oral bioavailability of 5-ASA ranged from 19% for ileocecal release to 75% for release in the upper gastrointestinal tract, and urinary excretion of 5-ASA and Ac-5ASA over 48 h was 52%, 55%, and 21% of the dose, respectively, in which the total recovery after intravenous administration was 78% [120]. Due to the low permeability and low solubility of 5-ASA, 5-ASA was classified as Biopharmaceutical Classification System (BCS) Class IV with low solubility and low permeability [110].

## 6. Diagnosis of SIBO by Urinary Excretion Tests of Bile Acid Conjugates

Urinary excretion tests using bile acid conjugates for SIBO diagnosis were developed. Developed bile acid conjugates are hydrolyzed by bacterial bile acid (or cholylglycine) hydrolase in the intestinal lumen after oral administration and the split-conjugated moieties (PABA, 5ASA) are absorbed into the membrane and a part of them is metabolized by NAT in the intestine and liver and then into the urine. Enzyme-labile substrates consisting of PABA, or 5-ASA, and a bile acid developed were PABA-CA, PABA-UDCA, PABA-UDCA disulphate, and 5-ASA-UDCA monophosphate. 5-ASA-chenodeoxycholic acid (5-ASA-CDCA) was also examined, and this compound was mainly used as a colonic-targeting delivery system of 5-ASA and UDCA to treat IBD.

### 6.1. Use of PABA-CA

PABA-CA was developed to diagnose SIBO for laboratory animals (rats), in which rat models with bacterial overgrowth or contamination of fecal materials in the upper small intestine were prepared by surgical manipulation. In an in vitro study, PABA-CA was hydrolyzed in the presence of acetone powder of *Clostridium welchii* exhibiting bile salt hydrolase activity in pH 5.8 phosphate buffer. When 10 mg PABA-CA was administered orally to normal rats, antibiotics-treated rats (a mixture of tetracycline, ampicillin, and neomycin), and rats with experimental bacterial overgrowth, the average cumulative urinary excretion of PABA in 6 h after administration was 0.42 mg, 0.21 mg (50% decrease of control), and 1.11 mg (2.5-fold of control), respectively [93]. Healthy volunteers and patients with various gastrointestinal disorders ingested 1.2 g of PABA-CA along with 500 mL of orange juice after an overnight fast, and the efficacy of the urinary excretion test using PABA-CA was compared with that of ^14^C-D-xylose BT. The urinary excretion rates of PABA after 4 h in the ^14^C-D-xylose BT-positive group were 1.6% of the dose, which was significantly higher than those in the ^14^C-D-xylose BT-negative group and the healthy control group in which PABA excretion rates of both groups were approximately 0.7% of the dose. The agreement of SIBO diagnosis between the ^14^C-D-xylose BT and the urinary PABA-CA test was 85.7% (*p* < 0.01) [121]. These results suggest that the urinary PABA-CA test could be sensitive to detecting SIBO, as well as the case of ^14^C-D-xylose BT.

### 6.2. Use of PABA-UDCA

PABA-UDCA was developed for SIBO diagnosis [122,123]. PABA-UDCA was cleaved by various intestinal aerobic and anaerobic bacteria; (aerobic bacteria) *Enterococcus faecalis*, *Lactobacillus acidophilus*, *Proteus mirabilis*, and *Staphylococcus epidermidis*; (anaerobic bacteria) *Bacteroides fragilis*, *Bacteroides thetaiotaomicron*, *Bacteroides vulgatus*, *Bifidobacterium adolescentis*, *Bifidobacterium longum*, *Clostridium perfringens*, *Eubacterium aerofaciens*, and *Fusobacterium varium*, indicating that these bacteria exhibit bile salt (or cholylglycine) hydrolase activity. PABA-UDCA was not hydrolyzed by pancreatic enzymes, although other UDCA conjugates, such as L-leucine-UDCA and L-lysine-UDCA, were cleaved by pancreatic enzymes, carboxypeptidase, and intestinal bacteria acting as bile salt hydrolase. For comparison, GCA was hydrolyzed to CA completely by bile salt hydrolase and partly by pancreatic enzymes and carboxypeptidase A and B. In the rat’s everted intestine, PABA-UDCA was absorbed by active transport in the rat terminal ileum (in which the concentration ratio of PABA-UDCA in the serosal/mucosa side of the everted sac was approximately 5 under steady-state), but in other segments of the everted intestine, such as the upper and lower jejunum, upper ileum, cecum, and colon, PABA-UDCA was absorbed by passive diffusion (serosal/mucosal concentration ratio was approximately 1 under steady-state conditions) [122]. When PABA-UDCA (10 mg) was administered orally to normal rats, rats treated with antibiotics (a mixture of polymyxin B and tinidazole), and rats with experimental bacterial overgrowth induced by enteric blind loop, the average cumulative urinary excretion of PABA 6 h after administration was 0.339 mg, 0.014 mg (96% decrease of control), and 0.674 mg (2.0-fold of control), respectively. It was commented that these results indicated that PABA-UDCA offers a simple and rapid method for diagnosing SIBO [123]. Additionally, bactericidal potencies of various antibiotics were evaluated by using PABA-UDCA in rats with intestinal bacterial overgrowth due to enteric stagnant loops. When untreated, normal rats received PABA-UDCA at a dose of 10 mg, the urinary excretion rate of PABA in 6 h after administration was 0.339 mg, but in rats treated with antibiotics, the urinary excretion rate of PABA significantly decreased as follows: treatment with ampicillin, doxycycline, and fradiomycin, PABA excretion was 0.0183 mg, 97.3% decrease of untreated rat; treatment with polymyxin B and tinidazole, 0.0140 mg, 97.9% decrease; treatment with polymyxin B, 0.225 mg, 66.6% decrease; treatment with tinidazole, 0.0427 mg, 93.7%; treatment with kanamycin, 0.0503 mg, 92.5% decrease [124]. The clinical usefulness of PABA-UDCA was evaluated in human volunteers. When 250 mg PABA-UDCA was ingested, the average amounts of PABA excreted into the urine during the 6 h after the dosing was 0.0211 mg in controls and 0.0122 mg (42.2% of control) in the group treated with polymyxin B and tinidazole, respectively. In these studies, no adverse effect was observed in human volunteers [125]. These results indicate that the urinary excretion test of PABA-UDCA is useful for SIBO diagnosis.

### 6.3. Use of PABA-UDCA Disulfate

The usefulness of PABA-UDCA disulphate, a hydrophilic salt of PABA-UDCA, was evaluated. This compound was not absorbed from any part of the small intestine in the everted intestinal sac, and the biliary excretion of PABA after intra-ileal instillation of PABA-UDCA disulphate was negligible in rats, indicating that intact PABA-UDCA disulphate is an unabsorbable single-pass type of drug. Since PABA-UDCA disulfate is rapidly hydrolyzed by bacterial bile salt hydrolase in the small intestine, only the split PABA and UDCA are absorbed into the membrane [122,123]. When PABA-UDCA disulfate was administered to rats at a dose of 15 mg, control rats excreted 0.188 mg of PABA, rats with an intestinal stagnant loop exhibiting intestinal bacteria overgrowth excreted 0.530 mg (2.82-fold of control), and rats treated with antibiotics (polymyxin B and tinidazole) excreted 0.0049 mg (2.6% of control and less than 1% of rats with bacterial overgrowth), kanamycin-treatment 0.0310 mg (16.5% of control and 5.8% of rats with bacterial overgrowth rats), and clindamycin-treatment 0.0409 mg (21.7% of control and 7.7% of rats with bacterial overgrowth) during the 6 h after the oral administration [126].

In a clinical setting, the H_2_ BT with 25 g lactose and/or 10 g lactulose and urinary PABA-UDCA (250 mg dose) test were performed simultaneously on 46 patients with suspected contaminated small bowel syndrome and on nine healthy subjects. A total of 10 out of 25 patients exhibited pathologic higher urinary PABA excretion (12.77 mg of PABA) compared to the control (4.1 mg of PABA), and in nine out of the same group exhibited positive H_2_ BT (early rise of >20 ppm of H_2_ gases), while in six cases both tests proved to be pathological. Treatment with tinidazole, an antibiotic that is used to treat certain types of vaginal infections, for 10 days significantly decreased the PABA excretion in SIBO patients [127]. The usefulness of the urinary PABA-UDCA test was further evaluated by expanding participants (68 patients with suspected contaminated small bowel syndrome, and 10 healthy control subjects). In 13 of these 35 patients, the urinary PABA excretion increased from 3.6 mg of control to 11.7 mg (3,25-fold of control), indicating bacterial overgrowth, 15 of the 35 patients gave a positive H_2_ BT, and in the remaining seven patients showed abnormality in both tests. In eight patients, the urinary excretion of PABA was decreased significantly following a 10-day tinidazole treatment to 5.5 mg (42% of control) from 13.1 mg PABA excretion. Based on these results, it was concluded that the urinary PABA-UDCA test is a valuable clinical method for the detection of bacterial overgrowth, especially in cases where H_2_ production alone fails to reveal SIBO. Additionally, it was stated that this test is a useful procedure for evaluating the efficacy of antibacterial treatments [128].

### 6.4. Use of 5-ASA-bile acid Conjugates

5-ASA-UDCA monophosphate was developed, in which 5-ASA was used as an anti-inflammatory agent for the treatment of IBD. 5-ASA-UDCA monophosphate was hydrolyzed by bile salt (cholylglycine) hydrolase and released 5-ASA, but it was not hydrolyzed by pancreatic and intestinal mucosal enzymes. Unlike PABA-UDCA, 5-ASA-UDCA monophosphate was not absorbed by the small intestine. When 5-ASA-UDCA monophosphate was administered intravenously at a dose of 30 mg in rats, this compound was excreted into bile and urine at a rate of 48.5% and 1.7% of the dose, respectively, in 22 h after injection. When 5-ASA-UDCA monophosphate was administered orally, 5-ASA-UDCA monophosphate was not detected in the bile and urine, indicating that the intestinal absorption of 5-ASA-UDCA monophosphate itself is negligible. In contrast, however, N-acetyl-5-ASA (Ac-5ASA), a metabolite of 5-ASA produced by NAT, was observed in the urine after oral administration of 5-ASA-UDCA monophosphate. Urinary excretions of Ac-5ASA were measured for 24 h following the oral administration of 20 mg of 5-ASA-UDCA monophosphate, control rats excreted 276.3 µg of Ac-5ASA and the rats with intestinal stagnant loop (SIBO) excreted 1224.1 µg of Ac-5ASA (4.43-fold of control) in the urine [129,130]. Separately from the SIBO diagnosis, 5-ASA-UDCA can facilitate the delivery of both 5-ASA and UDCA to the colon, which would be useful for the treatment of colonic IBD such as ulcerative colitis and Crohn’s disease. When 5-ASA-UDCA was administered to rats, this compound was not absorbed from the duodenum but was concentrated in the colon where it was partially hydrolyzed by the intestinal bacteria to UDCA and 5-ASA. Similarly, 5-ASA-UDCA and 5-ASA-chenodeoxycholic acid (5-ASA-CDCA) were administered orally to deliver 5-ASA to the large intestine, in which 5-ASA-CDCA was newly developed [131]. Both 5-ASA-bile acid conjugates were sufficiently delivered to the large intestine without deconjugation and the lower dose of 5-ASA-CDCA was enough for the treatment of ulcerative colitis in colonic IBD [132]. The inhibitory effect of 5-ASA-UDCA against colon carcinogenesis was examined in rats, in which colon carcinogenesis was induced by intrarectal instillation of 2 mg of N-methyl nitrosourea three times a week for 3 weeks. The tumor incidence and the mean number of tumors/rats at week 30 were significantly lower and smaller in the 5-ASA-UDCA-containing diet (0.11% and 0.02%) groups in the following way: 48% and 0.7 in the 5-ASA-UDCA diet group compared to the control group, 83%, and 1.3 [133]. These results may suggest that 5-ASA and UDCA produced from UDCA-5-ASA are effective in the treatment of colonic IBD and carcinogenesis.

## 7. Treatment of SIBO

The treatment of SIBO is achieved using antibiotics, probiotics, and/or herbal medicine. In addition, the efficacy of fecal microbiota transplantation for SIBO treatment is also reported.

### 7.1. Treatment of SIBO with Antibiotics

SIBO is commonly treated with antibiotics, such as cephalexin, chloramphenicol, ciprofloxacin, doxycycline, metronidazole, neomycin, norfloxacin, rifaximin, tetracycline, a mixture of amoxicillin and clavulanate, a mixture of amoxicillin and ciprofloxacin and metronidazole, or a mixture of trimethoprim and sulfamethoxazole [134,135,136]. The eradication of SIBO with antibiotics is the first-line treatment to provide symptom relief. Limited numbers of controlled studies have shown systemic antibiotics to be efficacious. In contrast, however, many researchers reported the effectiveness of rifaximin against SIBO. Rifaximin is a gastrointestinal-selective antibiotic that, due to its lack of systemic absorption with a broad spectrum of antimicrobial activity, has an excellent safety profile and a negligible impact on the intestinal microbiome. This agent has been used for treating travelers’ diarrhea, as well as other gastrointestinal diseases [45,137]. In a review of clinical trials with rifaximin, this agent improved global symptoms in 33–92% of patients and eradicated SIBO in up to 84% of patients with IBS, with results sustained for up to 10 weeks post-treatment [138]. Proton pump inhibitors (PPIs) are known to cause diarrhea, enteric infections, and alter the gastrointestinal bacterial population by suppressing the gastric acid barrier, and SIBO was detected in 50% of patients using PPIs, 24.5% of patients with IBS, and 6% of healthy control subjects. Therapy with rifaximin (400 mg three times per day for 2 weeks) eradicated 87–91% of the cases of SIBO in patients who continued PPI therapy [139]. The efficacy, safety, and tolerability of rifaximin 1600 mg for 7 days (Group 1) with respect to 1200 mg/day for 7 days (Group 2) for SIBO treatment were assessed. Group 1 showed significantly higher efficacy than Group 2 for SIBO treatment (80–82% vs. 58–61%) with similar compliance and side-effect profile [140]. SIBO commonly reoccurs even after successful eradication with antibiotics. For example, in patients with SIBO remission after treatment with rifaximin (1200 mg per day for 1 week), a total of 12.6%, 27.5%, and 43.7% of patients showed positivity to glucose BT at 3, 6, and 9 months after successful antibiotic treatment, respectively. Older age, a history of appendectomy, and chronic use of PPIs were found to be associated with glucose BT positivity recurrence [141]. To prevent the recurrence of IBS symptoms after successful antibiotic treatment, the effect of low-dose erythromycin (50 mg) and low-dose tegaserod (2–6 mg) orally at bedtime was examined. Subjects receiving no prevention showed symptom recurrence after 59.7 days on average. Prevention using erythromycin demonstrated 138.5 symptom-free days compared to 241.6 days with tegaserod. Switching from erythromycin to tegaserod extended resolution from 105.8 days to 199.7 days [142]. These data would indicate that SIBO is likely to recur sooner or later. Alternative therapies, such as probiotics, therapeutic diets, and herbal medicines, have been used to individualize SIBO management, particularly in recalcitrant cases. Probiotics are found in food, yogurt, or oral capsules containing live bacteria and/or yeasts such as *Lactobacillus*, *Bifidobacterium*, and *Saccharomyces boulardii.* After an initial 3-week therapy with broad-spectrum antibiotics, a 15-day maintenance antibiotic therapy with lactol (a combination of probiotic, Lactobacillus sporogeneses, and prebiotic, fructo-oligosaccharides) was administered to assess the efficacy of a probiotic consisting of lactobacilli in the treatment of SIBO. The H_2_ BT turned negative in 93.3% of those receiving lactol compared to 66.7% of the controls. In all the cases receiving lactol, the abdominal pain disappeared completely. These results indicated that adding probiotics to the maintenance therapy of SIBO patients on routine antibiotic therapy could be beneficial in preventing the complications of this syndrome [143].

### 7.2. Treatment of SIBO with Probiotics

The efficacy of probiotics treatment (Lactolevure^®^, twice/day for 30 days) in the improvement of symptoms of IBS was evaluated in patients with SIBO. Thirty days after the end of treatment, a 71.3% decrease in the total IBS score was detected in patients with IBS and SIBO compared to 10.6% in those without SIBO. The benefit of probiotics was greater among patients with a pro-inflammatory cytokine pattern in the duodenal fluid [144]. Separately, the efficacy of probiotics was analyzed by using published full-text articles and abstracts, and it was found that probiotics supplementation could effectively decontaminate SIBO, decrease H_2_ concentration, and relieve abdominal pain, but were ineffective in preventing SIBO [145]. Dietary strategies for the treatment of SIBO are based on a reduction in the consumption of fermentable products, which involves a diet low in fiber, sugar alcohols and other fermentable sweeteners such as sucralose. A diet low in fermentable includes oligosaccharides, disaccharides, monosaccharides, and polyols, which are short-chain carbohydrates that are osmotically active and fermentable by small intestinal bacteria. It was stated that there is still not enough scientific evidence to support a specific type of diet for the treatment of SIBO [136].

### 7.3. Treatment of SIBO with Herbal Medicine

A number of herbs, such as garlic, black cumin, cloves, cinnamon, thyme, all-spices, bay leaves, mustard, and rosemary, exhibit antimicrobial properties, and these herbs may be available in SIBO treatment [146]. Subjects with newly diagnosed SIBO by lactulose BT were given two capsules of the following commercial herbal preparation (twice daily for 4 weeks); Dysbiocide and FC Cidal (Biotics Research Laboratories, Rosenberg, TX, USA) or Candibactin-AR and Candibactin-BR (Metagenics, Inc, Aliso Viejo, CA, USA). Herbals appeared to be as effective as antibiotic therapy (rifaximin, 1200 mg per day for 4 weeks) for SIBO rescue therapy for rifaximin non-responders, and it was concluded that prospective studies are needed to validate these findings and explore additional alternative therapies in patients with refractory SIBO [134]. Collectively, it was stated that large-scale, randomized, placebo-controlled trials are needed to further evaluate the best way to utilize alternative therapies such as probiotics, therapeutic diets, and herbal medicines in the treatment of SIBO [147].

### 7.4. Treatment of SIBO with Fecal Microbiota Transplantation

Fecal microbiota transplantation (FMT) is also used for SIBO patients because FMT is an effective tool for reestablishing the structure of gut microbiota. For example, when the effect of donor/recipient SIBO status on FMT outcomes and post-FMT gastrointestinal symptoms was examined, there was a trend toward increased gastrointestinal symptoms among recipients receiving stool from lactulose H_2_ BT-positive (SIBO) donors [148]. Additionally, SIBO patients diagnosed with lactulose H_2_ BT were randomized into two groups and were given FMT capsules containing healthy microbiota or placebo capsules once a week for 4 consecutive weeks. Gastrointestinal symptoms significantly improved in SIBO patients, and the gut microbiota diversity significantly increased after FMT treatment compared to participants in the placebo group, suggesting the usefulness of FMT in treating SIBO [149].

## 8. Discussion

In this article, we reviewed SIBO diagnostic tests including cultural analysis of bacteria in duodenum/jejunum fluid aspirates, BTs of H_2_/CH_4_/CO_2_ gases after ingestion of carbohydrates and/or lipids, and urinary excretion tests of bile acid conjugates with PABA or 5-ASA (Table 1). Among these diagnostic methods of SIBO, the consensus and/or guidelines of diagnostic procedure and criteria (cut-off levels) are determined for cultural analysis of duodenum/jejunum fluid aspirates and/or BTs using hydrocarbons in various countries or regional communities, for example, such as Italy, Rome [19], North America [10], Europe [22], pan-Europe [65], and Asian-Pacific [21]. In general, the cultural analysis of duodenum/jejunum fluid aspirates is accepted as the gold standard in diagnosing SIBO. This method, however, is also reported to be high-cost, invasive, and not without risk to the patients. It is difficult to perform cultural analysis of bacteria in a patient’s intestinal fluid without causing cross-contamination in the daily clinical routine because a good aseptic technique is required [5,43,44,45,46,150]. In summary, the results of the culture of jejunal fluid aspirates could potentially be affected by cross-contamination from oropharyngeal and luminal microbes, and there may be controversy regarding the best cut-off values for SIBO diagnosis. Cultural analysis of aspirates is rarely used in routine clinical settings, and these limitations have led to the development of BTs [36].

BTs have sub-optimal sensitivity and specificity for SIBO diagnosis, as compared with the cultural analysis of aspirates [36]. Many researchers pointed out the shortcomings of BTs and the need for further improvement of standardization and validation for SIBO because BTs are indirect methods involving a sequence of reactions and metabolic pools, and they usually supply semiquantitative data, in which H_2_/CH_4_/CO_2_ gases are produced not only from diagnostic substrates ingested but also from daily dietary food and metabolism of endogenous substances [45,151]. Among BTs, H_2_ lactulose and glucose BT were recommended as the most utilized diagnostic methods of SIBO in the clinical setting [5,43]. In contrast, however, other researchers reported that there was no significant correlation between glucose BT results with either the number of bacterial colonies or the DNA-based bacterial cell counts [40,44,46]. In addition, it was reported that lactose BT should be neither recommended nor suggested to detect SIBO in clinical practice, and glucose BT remains the most accurate BT for the non-invasive diagnosis of SIBO despite its low sensitivity [152]. Similarly, it was reported that the lactulose BT does not perform well in the following ways: there is poor agreement between lactulose BTs and cultural analysis of duodenal aspiration in evaluating SIBO [40]. Additionally, it was reported that the lactulose BT is based upon an incorrect premise, and therefore, incorrect interpretations, resulting in the over-diagnosis of SIBO. They concluded that the lactulose BT should be discarded for future use [20]. In contrast, the combined use of H_2_ and CH_4_ BTs was proposed as follows: the H_2_-based lactose malabsorption test resulted in about 5–15% false negatives mainly due to CH_4_ production by methanogenic archaea, and the combined measurement of H_2_ and CH_4_ can offer considerable improvement in the diagnosis of malabsorption syndromes and SIBO when compared with a single H_2_ BT [7,55,61,153]. The measurement of CH_4_ in addition to H_2_ can increase the sensitivity of BT for SIBO, and diagnostic accuracy of H_2_ BT for SIBO can be maximized by careful patient selection for testing, proper test preparation, and standardization of test performance, as well as test interpretation [154]. It was reported that compared to the cultural analysis of jejunal aspirates by 16S rRNA gene sequencing, the glucose-based H_2_ and CH_4_ BTs were not sensitive to the overgrowth of jejunal bacteria. However, a positive BT may indicate altered jejunal function and microbial dysbiosis [44]. For example, in newly diagnosed chronic pancreatitis patients, the jejunal aspirate culture was positive in 37.5% of patients, while the glucose H_2_ BT showed that 29% of patients had SIBO. It was concluded that glucose H_2_ BT has low sensitivity but had high specificity in the diagnosis of SIBO in chronic pancreatitis [155]. Taken together, it may be concluded that careful attention, preparation, further standardization and validation for the methodology and interpretation of BT results are necessary to improve SIBO diagnosis [7,10,45].

As urinary excretion tests for SIBO diagnosis, PABA-CA, PABA-UDCA, PABA-UDCA disulfate, and 5-ASA-UDCA monophosphate were developed as diagnostic substrates [93,121,122,123,128], in which PABA-UDCA has been proved to be useful in diagnosing SIBO in humans, without adverse clinical events [125,127,128]. In the intestinal lumen, all these bile acid conjugates are hydrolyzed by various intestinal aerobic and anaerobic bacterial bile salt (cholylglycine) hydrolase, but not by other enzymes, such as pancreatin, carboxypeptidase, human plasma, intestinal mucosa homogenates and liver homogenates. The released PABA and 5-ASA are absorbed by passive diffusion, although 5-ASA was reported to be absorbed in a saturable manner at a low concentration, and dominated by passive transcellular and paracellular processes at higher concentrations [110]. Orally ingested PABA-UDCA is also absorbed as it is by passive diffusion along the small intestine except for the terminal ileum, and actively in the terminal ileum [122]. In contrast, both PABA-UDCA disulfate and 5-ASA-UDCA monophosphate were not absorbed throughout the small intestine possibly due to their high hydrophilicity [124,129]. The intestinal absorption of PABA, produced by bacterial enzymatic hydrolysis, was rapid with a peak time of less than 60 min after ingestion, and the absorption rate was almost 100% of the produced PABA amount and mainly metabolized to Ac-PABA by NAT in the intestinal membrane and liver. Ac-PABA is also excreted into the urine [99,101,102]. The total recovery of PABA and Ac-PABA in the urine was more than 85% (85–110%) of the dose [100]. These pharmacokinetic properties of PABA, PABA-UDCA and PABA-UDCA disulfate would suggest that the activity of bacterial bile salt hydrolase can be detected at high sensitivity by determining the urinary PABA excretion rate released from PABA-UDCA in the proximal intestine in the early stage (around 5 h) after PABA-UDCA ingestion. The contribution of endogenous PABA-related substances in the urine, if there were any, on the diagnostic value could be cancelled by subtracting the concentration of PABA-related substances in the control baseline urine (preposing stage). The pharmacokinetic properties of UDCA conjugates (PABA-UDCA, PABA-UDCA disulfate, and 5-ASA-UDCA monophosphate) developed for SIBO diagnosis and its conjugated moieties (PABA, 5-ASA) are summarized in Table 2.

5-ASA, the first-line drug for IBD treatment, is reported to be rapidly absorbed in the small intestine, and a part of 5-ASA is metabolized to Ac-5ASA by NAT [153,155]. For example, when 5-ASA (1 g) was administered orally, the peak concentrations of 5-ASA and Ac-5ASA were observed at 1 h after ingestion [156,158]. Regional perfusions of 5-ASA in the small intestine of anaesthetized rats resulted in the appearance of Ac-5ASA in the intestinal lumen, and the luminal appearance of Ac-5ASA was five-fold higher in the jejunum than in the ileum. Ac-5ASA significantly decreases 5-ASA absorption at low luminal drug concentrations in Caco-2 cells, suggesting that both 5-ASA and Ac-5ASA are absorbed via the same influx transporter [114]. The apical efflux of Ac-5ASA was reduced and the Ac-5ASA cellular accumulation, was increased in the presence of MK571 and indomethacin, both are inhibitors of multidrug resistance-associated proteins (MRPs). This indicates that Ac-5ASA is a substrate of MRPs and the luminal efflux of Ac-5ASA is mediated by MRP2 [114]. The expression of mRNA for MRP is highest in the jejunum and decreased toward more distal regions in the human intestine [159]. Additionally, the rat jejunum exhibited a higher apical MRP2 expression and a lower basolateral MRP3 expression, and the ileum exhibited a lower apical MRP2 and a higher basolateral MRP3 [160]. Regarding the membrane permeability of 5-ASA, SMCT1, an influx transporter, has been reported to accept 5-ASA as a substrate, and SMCT1 is expressed in the large intestine [116]. When healthy volunteers received 1.5 g of 5-ASA tablet with sustained release per day for 6 days, 40% of the 24 h dose was recovered from feces and 53% from the urine at a steady state [160]. When human volunteers ingested 1.0 g 5-ASA suspension, 78.3% of the dose (21.2% 5-ASA, 57.1% Ac-5ASA) was excreted in the urine, and 11.3% of the dose was eliminated in the feces, consisting mostly of Ac-5ASA. Food coadministration reduced 5-ASA and Ac-5ASA bioavailability to 9.3%% and 43.4%, respectively, reduced the salicylates’ urinary excretion to 46.8% and increased fecal salicylate elimination to 24.2% of the total dose [156]. When 5-ASA suspension (800 mg) was administered to healthy volunteers, patients with ulcerative colitis, and patients with Crohn’s disease in remission, systemic absorption of 5-ASA and Ac-5ASA was low (about 20% of dose) and did not differ between the three groups, and more than 80% of the drug was found in the terminal ileum and colon [118,161]. When 5-ASA (1 g) was administered as Pentasa^®^-controlled release capsules to healthy human volunteers, 29.4% of the dose was excreted in the urine primarily as Ac-5ASA, and 40% of the dose was eliminated in the feces [157]. Active absorption of 5-ASA-UDCA monophosphate was not observed even at the terminal ileum [129].

Based on the pharmacokinetic properties, the fate of PABA-UDCA and 5-ASA-UDCA monophosphate in the proximal small intestine is presented in Figure 2.

Under SIBO, the number of bacteria in the upper gastrointestinal tract is increased and the type of bacteria could be altered. Orally administered PABA-UDCA and 5-ASA-UDCA monophosphate are hydrolyzed by aerobic and/or anaerobic bacterial bile salt hydrolase, and PABA and 5-ASA are released, respectively. Both PABA and 5-ASA are metabolized partly to Ac-PABA and Ac-5ASA, respectively, and are excreted into the urine. The state of SIBO will produce PABA and 5-ASA more rapidly and at a higher rate in the proximal small intestine. PABA is absorbed by passive diffusion via transcellular and possibly paracellular routes and 5-ASA is absorbed via transporter-mediated transport at a low concentration and by passive diffusion after the saturation of transporter-mediated transport at a higher concentration into the intestinal membrane. In the intestinal membrane, both PABA and 5-ASA are metabolized partly to N-acetyl metabolites by NAT and then in the liver. In the case of PABA conjugates, the produced PABA in the intestinal lumen is absorbed and partly metabolized into Ac-PABA in the intestinal membrane and liver. Both unmetabolized PABA and Ac-PABA are excreted into the urine at a rate of almost 100%. The intact PABA-UDCA itself and UDCA generated in the proximal intestine are also absorbed by passive diffusion, transported into the portal vein, absorbed by the liver, excreted into the bile, and subjected to enterohepatic circulation. During this enterohepatic circulation, UDCA is partly conjugated with glycine and/or taurine (GUDCA and/or TUDCA) in the liver. In the case of 5-ASA-UDCA monophosphate, the generated 5-ASA by intestinal bacterial bile salt hydrolase is absorbed and metabolized to N-acetyl-5ASA (Ac-5ASA) by N-acetyltransferase in the intestinal membrane. In contrast to the case of PABA-UDCA, the produced UDCA-monophosphate and intact 5-ASA-UDCA monophosphate are not absorbed possibly due to the low lipophilicity. Ac-5ASA is a substrate of ATP-dependent efflux transporter MRPs and could be effluxed mostly into the intestinal lumen by apical MRP2-mediated transport or into the portal vein by basolateral MRP3-mediated transport because the rat jejunum exhibited a higher apical MRP2 and a lower basolateral MRP3 expression than the ileum [156]. Thus, different from the case of PABA-UDCA and PABA, it may be considered that the urinary excretion rates of 5-ASA and Ac-5ASA could be varied depending on the activities of NAT in the intestinal tissue, apical MRP2, basolateral MRP3, and also the regional difference of 5-ASA-UDCA hydrolysis in the small intestine. Collectively, in the case of 5-ASA-UDCA-monophosphate, this compound is considered to be useful and valuable as a delivery system of 5-ASA, the first-line drug for IBD treatment, to the colonic region, instead of SIBO diagnosis [130,131].

## 9. Conclusions

SIBO is generally treated with antibiotics efficiently. Thus, to avoid unnecessary treatment with antibiotics, it is very important to diagnose SIBO correctly and rapidly. As discussed in Section 8, the urinary excretion test of PABA bile acid conjugated is thought to be a useful diagnostic test for SIBO because PABA-related compounds produced from PABA-UDCA by bacterial bile salt hydrolase in the small intestine and by intestinal NAT are excreted into urine efficiently (at a rate of almost 100%) and are easily determined by using an assay kit. The control baseline level of PABA-related compounds in the urine before ingestion of PABA-UDCA or contamination, is rapidly estimated and cancelled from the test value obtained after ingestion of PABA-UDCA in each patient. In addition, the necessary time for the urinary PABA-UDCA excretion test (from ingestion to urine sampling) is relatively short (about 4–6 h). Collectively, PABA-UDCA, especially PABA-UDCA disulfate, is likely to offer a simple and safe method for the evaluation of SIBO without the use of radioisotopes or expensive, special apparatus. It will be important to confirm the usefulness of PABA-UDCA disulfate as a substrate for SIBO diagnosis in patients. The N-acetyl metabolite of 5-ASA, or Ac-5ASA that is generated from 5-ASA-UDCA monophosphate by bacterial bile salts hydrolase is a substrate of MRPs, an ATP-dependent efflux transporter. In addition, 5-ASA is also a substrate of SMCT1, an influx transporter, indicating the intestinal absorption of 5-ASA is variable depending on the luminal concentration, or among hydrolysis rate of 5-AS-UDCA monophosphate. Thus, the intestinal absorption of 5-ASA and the urinary excretion of metabolite(s) of 5-ASA-related compounds, such as Ac-5ASA, could be varied among patients depending on the activity of SMCT1-mediated influx transport, NAC-mediated metabolism, and MRPs-mediated efflux transport. Collectively, in the case of 5-ASA-UDCA monophosphate, this compound would be suitable as the colonic delivery system of 5-ASA, the first-line drug for IBD treatment, and carcinogenesis, rather than for SIBO diagnosis.

## Figures and Tables

**Figure 1 antibiotics-12-00263-f001:**
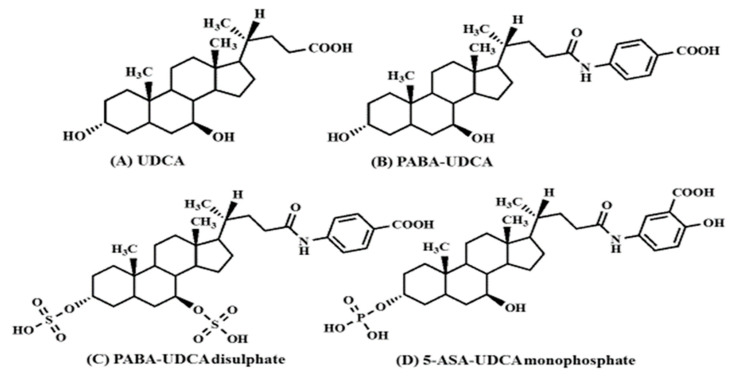
Chemical structures of (**A**) ursodeoxycholic acid (UDCA), (**B**) UDCA conju gated with para-aminobenzoic acid (PABA-UDCA), (**C**) UDCA disulfate conjugated with PABA (PABA-UDCA disulfate), and (**D**) UDCA monophosphate conjugated with 5-aminosalycilic acid (5-ASAUDCA monophosphate.

**Figure 2 antibiotics-12-00263-f002:**
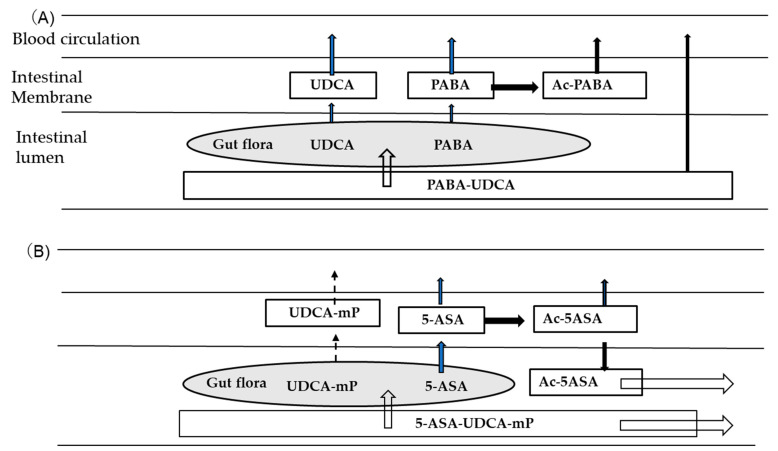
Schematic representation for the fate of (**A**) PABA-UDCA and (**B**) 5-ASA-UDCA mono-phosphate in the proximal small intestine. (**A**): A part of orally ingested PABA-UDPA is deconjugated by intestinal bacterial bile salt (cholylglycine) hydrolase. The released PABA and UDCA are absorbed by passive diffusion. A part of PABA is metabolized to N-acetyl-PABA (Ac-PABA) by N-acetyltransferase in the intestine and liver. These PABA and Ac-PABA are efficiently excreted into the urine. UDCA and a part of intact PABA-UDPA are absorbed by passive diffusion in the proximal intestine and UDCA is conjugated with taurine and/or glycine in the liver. These UDCA, UDCA-amino acids conjugates and PABA-UDCA are subjected to enterohepatic circulation. (**B**): A part of orally ingested 5-ASA-UDPA monophosphate (5-ASA-UDCA-mP) is deconjugated by intestinal bacterial bile salt (cholylglycine) hydrolase. The released 5-ASA is absorbed by carrier-mediated transport at a low concentration and passive diffusion at a higher concentration and is metabolized to N-acetyl-5-ASA (Ac-5ASA) by N-acetyltransferase. Ac-5ASA is effluxed into the intestinal lumen by apical MRP2 or into a portal vein by basolateral MRP3. Un-metabolized, 5-ASA and Ac-5ASA absorbed into blood circulation are excreted into the urine. Intact 5-ASA-UDCA monophosphate (5-ASA-mP) and UDCA-monophosphate (UDCA-mP) are hardly absorbed in the small intestine (single-pass type substance). Solid arrows represent the transport and/or metabolism of the compound in the intestinal membrane. The dotted arrows represent small transport in the membrane. Open arrows represent the metabolism and/or migration toward the distal intestinal lumen.

**Table 1 antibiotics-12-00263-t001:** Diagnostic tests for small intestinal bacterial overgrowth (SIBO).

Tests	Diagnostic Procedure	Refs.
Cultural analysis	A sampling of duodenum/jejunum fluid aspirates, the culture of bacteria in aspirates, and counting of colony-forming units (CFU) of bacteria. >10^3^ or ≥10^3^ CFU/mL are generally considered SIBO positive.	[7,10,41,44,47,51]
	Analysis of the mucosal microbiota composition by 16S ribosomal RNA (rRNA) gene sequencing for identification, DNA-based cell counting, and characterization.	
Breath Test (BT)	Detection of H_2_ and/or CH_4_ gases excreted in the breath after oral ingestion of a certain amount of carbohydrates. Consensus oral doses for lactulose, glucose, fructose, and lactose BT are 10, 75, 25, and 25 g, respectively. Positive lactulose or glucose BT for H_2_: rise above baseline ≥20 ppm by 90 min and bloating. Positive lactulose or glucose BT for CH_4_: ≥10 ppm at any point during testing and stool *M. smithii* (methane-producing organism). Carbon isotope-labelled (^13^C or ^14^C) D-xylitol, lipids, such as triolein and palmitic acid, glycocholate, and lactose-ureide (LU), were used as substrates of BT for SIBO diagnosis (detection of ^13^CO_2_ or ^14^CO_2_ gases). LUBT was also used to assess orocaecal transit time.	[7,10,56,64,70,71,150]
Urinary excretion tests	Oral administration of PABA-bile acid-conjugates; PABA-CA, PABA-UDCA, or PABA-UDCA disulfate, and urinary excretion rates of total PABA including N-acetyl-PABA that was split by bacterial-bile salt hydrolase are determined.	[121,122,123,124,125,126,130]
	Oral administration of 5-ASA-UDCA monophosphate and urinary excretion rate of N- acetyl-5-ASA, a metabolite, that was split by bacterial bile salt hydrolase is determined. In addition, 5-ASA-bile acid conjugates are effective in the delivery of both 5-ASA and UDCA to the colon to treat inflammatory bowel diseases.	

PABA, para-aminobenzoic acid; CA, cholic acid; UDCA, ursodeoxycholic acid; 5-ASA, 5-aminosalicylic acid.

**Table 2 antibiotics-12-00263-t002:** Pharmacokinetic properties of bile acid conjugates (PABA-UDCA, PABA-UDCA, 5- ASA-UDCA monophosphate) and its conjugated moieties (PABA, 5-ASA).

Compounds	Pharmacokinetic Properties	Ref.
PABA-UDCA	This compound is absorbed actively in the terminal ileum in rats. In other intestinal regions, active transport is not observed. This compound is cleaved by intestinal bacterial bile salt hydrolase and releases PABA, but not by other enzymes such as pancreatin, carboxypeptidase A/B, plasma, and intestinal/liver homogenates.	[122,123]
PABA-UDCA disulfate	This compound is not absorbed by the intestine of rats (a single-pass type substance). This compound is cleaved by intestinal bacterial bile salt hydrolase and releases PABA.	[124,125,126]
5-ASA-UDCA monophosphate	The intact compound is not absorbed by the small intestine of rats, cleaved by intestinal bacterial bile salt hydrolase, and releases PABA, but not by other enzymes such as pancreatin, carboxypeptidase, plasma, and intestinal/liver homogenates. After oral ingestion, the un-deconjugated fraction is delivered to the colonic region.	[129,130]
PABA	PABA is absorbed by passive diffusion in the small intestine at a rate of approximately 100% and total PABA, including its metabolites, is excreted into urine at approximately 100%. PABA is metabolized mostly to N-acetyl-PABA by arylamine N-acetyltransferase (NAT) in the liver and intestine in a saturable manner.	[101,102,156,157]
5-ASA	Transport of 5-ASA is saturable at low doses, dominated by passive, paracellular processes at higher doses in Caco-2 cells. 5-ASA is metabolized to N-acetyl-5-ASA, and N-acetyl-5-ASA is refluxed into the intestinal lumen by efflux transporter, MRP2, or excreted into urine. The elimination of the half-lives of 5-ASA and N-acetyl-5-ASA are short (0.5 to 1.5 h) and slow (5 to 10 h), respectively. 5-ASA is a BCS class IV drug with low solubility and low permeability.	[105,114,115]

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
