# Peer review of "Diagnosis by Microbial Culture, Breath Tests and Urinary Excretion Tests, and Treatments of Small Intestinal Bacterial Overgrowth"

_antibiotics, 2023, doi:10.3390/antibiotics12020263_

Round 1

Reviewer 1 Report

The title of the article (Diagnosis of Small Intestinal Bacterial Overgrowth using Bile Acid Conjugates) is not suitable because you included other diagnostic methods (BT, cultural analysis of duodenum/jejunum fluid aspirates). Please revise.

Describing the definition, etiology, and prevalence of SIBO using subtitles may be helpful.

“Interaction of endogenous/exogenous compounds with intestinal bacteria” Lines from Intestinal bacteria contain various………. To locally high concentrations of NSAID aglycones [25].

The article should be better organized using subheadings such as:

·       Definition

·       Clinical magnifications

·       Microbiology

·       Etiology

·       Pathophysiology and pathology

·       Pharmacokinetic properties

·       Diagnostics

·       Treatment

 What is the goal of describing the bioavailability of orally administered drugs and non-steroidal anti-inflammatory drugs (NSAID)? This may be described under the pathophysiology of SIBO.

Can you provide a small table to summarize and classify the diagnostic methods for SIBO, reviewed in this article?

Can you describe the methods used for obtaining duodenum/jejunum fluid aspirates?

What is the difference between section (5. Biological and pharmacokinetic properties of bile acid conjugates used for SIBO diagnosis) and (5.1. Biological and pharmacokinetic properties of bile acids)?

“Patients with pancreatic carcinoma and extrahepatic biliary drainage received 750 mg UDCA in three divided doses, and absorption of UDCA increased from 39.8 % at day 3 to 61.1% at day 10 of the administered dose, indicating an improvement of the absorption rate after the decrease of cholestasis by 53.7% [87]. Simi-larly, when patients who had a complete extrahepatic biliary obstruction caused by pan-creatic carcinoma but had no intestinal or liver disease received UDCA at a dose of 750 mg, the oral bioavailability was 55.1% [88].” What about SIBO? It is more helpful to focus on SIBO.

In your text (sections 5., 5.1, 5.2, 5.3), it is not clear how to diagnose SIBO using these bile acids.

You may summarize section 5 in the pathophysiology and pathology of SIBO.

Author Response

To Reviewer 1

Thank you very much for your reviewing our manuscript and valuable suggestions/indication.

We revised our manuscript according to your suggestion/indication.

Corrections were made in the revised manuscript in red.

  • The title of the article (Diagnosis of Small Intestinal Bacterial Overgrowth using Bile Acid Conjugates) is not suitable because you included other diagnostic methods (BT, cultural analysis of duodenum/jejunum fluid aspirates). Please revise.

The title of the manuscript was altered to “Diagnosis by microbial culture, breath tests and urinary excretion tests, and treatments of small intestinal bacterial overgrowth” (Page 1st)

  • The article should be better organized using subheadings such as: Definition, Clinical magnifications, Microbiology, Etiology, Pathophysiology and pathology, Pharmacokinetic properties, Diagnostics,Treatment

We are not medical doctors, and our fields of expertise are Biopharmaceutics (pharmacokinetics/ pharmacodynamics) (Murakami T) and Drug Design (Maeda Y), respectively, of Pharmaceutical Sciences. We prepared this manuscript from the viewpoint of the pharmacokinetics (PK) of diagnostic substrates and the modulation of PK by intestinal bacteria (SIBO) as an influencing factor. Abstract and introduction sections were revised to make clear the aim of our article from the viewpoint of pharmacokinetics (Page 2nd).

  • What is the goal of describing the bioavailability of orally administered drugs and non-steroidal anti-inflammatory drugs (NSAID)? This may be described under the pathophysiology of SIBO.

In pharmacotherapy, the pharmacokinetics (especially metabolism) of certain drugs can be modulated greatly under the SIBO state, as well as many other factors. Thus, to conduct effective pharmacotherapy for accompanying diseases, the rapid and reliable diagnosis method and treatment of SIBO will be necessary. Some comments were added to the text (Page 2).

  • Can you provide a small table to summarize and classify the diagnostic methods for SIBO, reviewed in this article?

The diagnostic methods are summarized in Table 1. Some comments were added for Table 1 in Page 4th.

  • Can you describe the methods used for obtaining duodenum/jejunum fluid aspirates?

A short sentence was added for the method by citing a reference regarding the duodenum/jejunum fluid aspirates (Page 3rd).

  • What is the difference between section (5. Biological and pharmacokinetic properties of bile acid conjugates used for SIBO diagnosis) and (5.1. Biological and pharmacokinetic properties of bile acids)?

The title of 5 was changed to “Development of bile acid conjugates for SIBO diagnosis”. Some sentences were corrected a little (Page 7).

  • “Patients with pancreatic carcinoma and extrahepatic biliary drainage received 750 mg UDCA in three divided doses, and absorption of UDCA increased from 39.8 % at day 3 to 61.1% at day 10 of the administered dose, indicating an improvement of the absorption rate after the decrease of cholestasis by 53.7% [87]. Similarly, when patients who had a complete extrahepatic biliary obstruction caused by pancreatic carcinoma but had no intestinal or liver disease received UDCA at a dose of 750 mg, the oral bioavailability was 55.1% [88].” What about SIBO? It is more helpful to focus on SIBO.In your text (sections 5., 5.1, 5.2, 5.3), it is not clear

These data show the pharmacokinetic (absorption) properties of UDCA. UDCA is absorbed by passive diffusion, but at a higher dose, the oral bioavailability of UDCA could be decreased due to the low solubility of UDCA. Similarly, under abnormal liver function (with less bile flow), the absorption of UDCA is decreased because of the fewer biosurfactants (bile salts). These data will help to understand and speculate the pharmacokinetics of UDCA and UDCA conjugates under various disease states including SIBO.

  • You may summarize section 5 in the pathophysiology and pathology of SIBO.

The data in Section 5 and the fate of bile acids-conjugates in the proximal intestine are summarized in Table 2 and Fig. 2 (Page 17-18).

Reviewer 2 Report

Maeda and Murakami et al describe a review looking at a variety of bile acid conjugates as diagnostic substrates in evaluating small intestinal bacterial overgrowth across various medium (such as fluid aspirate, urinary, fecal and breath). These may present a less-invasive method as options.

For the introduction, the authors may consider provide an overview in terms of the epidemiology and signs/symptoms of the condition-why is this a research question that needs to be addressed?

For the methods, the authors may consider describing how the search for the literature review was conducted (over what time period, what search terms and database was used).

For the discussion, how do the reviewed methods compare across the cultural analysis of duodenum/jejunum fluid aspirates which is regarded as the gold standard in terms of performance metrics? (such as concordance, positive/negative percentage agreement, sensitivity/specificity etc). This can be elaborated in greater detail (beyond Table 1) because it represents the crux of the manuscript
-breath testing may be less specific and subjected to the volatility of compounds in breath (what are some ways that can be used to circumvent the shortcoming, in terms of pre or post-analytical variables)
-are there are blood-based genetic sequencing (in addition to 16s RNA) that are available and can better pinpoint identity of the bacteria

How do the reviewed methods disentangle the causal relationship as some of these conjugates or metabolites may arise from non-overlapping biological pathway or alternative sources that may contribute? This can add strength to the discussion
-the gases may be indirectly produced by other sources
-patient-related factors such as diet may also influence the measured levels

Author Response

To Reviewer 2

Thank you very much for your valuable suggestion/indication. We corrected our manuscript according to your suggestion/indication. The title of the manuscript was changed to “Diagnosis by microbial culture, breath tests and urinary excretion tests, and treatments of small intestinal bacterial overgrowth” (Page 1st) Corrections were made in the revised manuscript in red.

  • For the introduction, the authors may consider provide an overview in terms of the epidemiology and signs/symptoms of the condition-why is this a research question that needs to be addressed?

We are not medical doctors, and our fields of expertise are Biopharmaceutics (pharmacokinetics/ pharmacodynamics) (Murakami T) and Drug Design (Maeda Y), respectively, Pharmaceutical Sciences. We prepared this manuscript from the viewpoint of the pharmacokinetics (PK) of diagnostic substrates and the modulation of PK by intestinal bacteria (SIBO) as an influencing factor. The abstract and introduction sections were revised to make clear the aim of our article from the viewpoint of pharmacokinetics (Page 2nd).

  • For the methods, the authors may consider describing how the search for the literature review was conducted (over what time period, what search terms and database was used).

The corresponding sentence was changed to “Literature was searched using PubMed, Google Scholar, Online search engines and appropriate keywords such as SIBO and diagnosis” (Page 2).

  • For the discussion, how do the reviewed methods compare across the cultural analysis of duodenum/jejunum fluid aspirates which is regarded as the gold standard in terms of performance metrics? (such as concordance, positive/negative percentage agreement, sensitivity/specificity etc). This can be elaborated in greater detail (beyond Table 1) because it represents the crux of the manuscript.

It is difficult to compare the reliability and/or efficacy of each diagnostic method because we don’t know the accuracy of data, including the diagnostic performance. The evaluation of each diagnostic method is scattered greatly among different researchers. Thus, we only cited the author’s comments in references and compared the agreement and sensitivity. Further study would be necessary to develop reliable and safe SIBO diagnosis methods under various disease states. A short comment was added to the conclusion section.

Cultural analysis is regarded as the gold standard in diagnosing SIBO, and the accuracy of other diagnostic tests is evaluated by comparing with the result of cultural analysis. However, this cultural analysis is high-cost, invasive, and can cause cross-contamination and not without risk to the patients. Some comment regarding the technique of aspiration was added by citing reference to the text (Page 3).

-breath testing may be less specific and subjected to the volatility of compounds in breath (what are some ways that can be used to circumvent the shortcoming, in terms of pre or post-analytical variables)

In general, under fasted condition, a time course of the rise of CO2/CH4 gases after ingestion in breath from the baseline (pre-) will be measured. I have no idea to circumvent the shortcoming.

-are there are blood-based genetic sequencing (in addition to 16s RNA) that are available and can better pinpoint identity of the bacteria

Regarding the detection of bacterial 16S rRNA/16S rDNA gene fragments in plasma, some comments were added in the text by citing 2 preferences newly (Page 11).

How do the reviewed methods disentangle the causal relationship as some of these conjugates or metabolites may arise from non-overlapping biological pathway or alternative sources that may contribute? This can add strength to the discussion.

To diagnose SIBO,
-the gases may be indirectly produced by other sources
-patient-related factors such as diet may also influence the measured levels

In the case of the urinary excretion test of bile acid conjugates, the contribution of unknown substances in the sample urine, if there were, would be cancelled by determining the concentration in control urine (predosing stage), as well as the case of BT. These comments were added to the discussion and conclusion sections (Page 17, 19).

Reviewer 3 Report

The article is very interesting before, it could be considered for further publication I have some major quires which authors need to incorporate and extensively revise their Manuscript.

General comments Font size needs to be cross checked. Grammatical mistakes need to be minimized. Sentence formation needs crosscheck.

Abstract section does not give proper information. Abstract means a full-fledged summary that should give readers highlights of the information and topics covered in the manuscript. Please revise the abstract (needs to rephrase and rewrite some sentences). Also, highlight essentialities and safety of isotope labeled moieties and future perspectives of the study.

Section Introduction

The authors are advised to avoid writing bold statements which don’t have reference to literature (e.g., The oral cavity contains more than 300 bacterial species, and the lower intestine contains about 500 bacterial species).

Avoid writing larger sentences such as “the extent of oral bioavailability of orally administered drugs is determined by the following three factors: the extent of membrane permeability (or extent of intestinal absorption), the extent of intestinal first-pass metabolism (or intestinal availability), and the hepatic first-pass metabolism (or hepatic availability). However, for drugs that undergo phase-2 conjugate metabolism and enterohepatic circulation, metabolism by intestinal bacteria is also involved” that too without proper citation to literature”

Supplement heading 2 with more studies regarding interaction of exo- and endogenous substances with bacteria inhabiting GI tract.

Rephrase and rewrite the starting information under heading 3 i.e., In clinical, to diagnose SIBO with an increased number and/or altered type of bacteria in the upper gastrointestinal tract, the quantitative culture of duodenum/jejunum fluid aspirates is conducted.

Take proper care of citation in the text e.g., [12.34]

Supplement sub-heading 3.1 with new information regarding cultural analysis at start of the sentence. Also, include some aspects of collecting the duodenum/jejunum aspirate.

Additionally, rephrase the statement “Increases in Gammaproteobacterial………” in sub-heading 3.2

In 3.3 sub0heading information is pertaining to 16SrRNA and in 3.4, there is sentence regarding 16SrDNA…. Please clarify

In sub-heading 3.3, please repharse the sentence and try to write in 2 sentences so that the meaning gets clarity.

In heading 7, there is information in one para which is too lengthy to get the desired meaning an d information. Please repharse the sentence and try to write in 2 or 3 sentences so that the meaning gets clarity.

The same problems persists in other sections as well. It is advised to revise the manuscript thoroughly and supplement different sections of the paper with updated information and proper reference to literature.

Section conclusion

The section needs to be a bit more elaborative and should highlights importance of the study and future directions with possible limitations.

My other suggestion is supplement the paper with some more tables and appropriate figures.

Author Response

To Reviewer 3

Thank you very much for your valuable suggestion/ indication. We corrected our manuscript according to your suggestion/indication. Corrections were made in the revised manuscript in red.

The title of the manuscript was also changed to “Diagnosis by microbial culture, breath tests and urinary excretion tests, and treatments of small intestinal bacterial overgrowth” (Page 1st)

  • General comments Font size needs to be cross-checked. Grammatical mistakes need to be minimized. Sentence formation needs crosscheck.

The font size was corrected. I tried to correct Grammatical mistakes, though it may not enough.

  • Abstract section does not give proper information.

By considering your suggestion, Abstract was revised greatly. 

  • Section Introduction; The authors are advised to avoid writing bold statements which don’t have reference to literature (e.g., The oral cavity contains more than 300 bacterial species, and the lower intestine contains about 500 bacterial species).

The comments on the oral cavity were made according to the reports of Ref. 2.

  • Avoid writing larger sentences such as “the extent of oral bioavailability of orally administered drugs is determined by the following three factors: the extent of membrane permeability (or extent of intestinal absorption), the extent of intestinal first-pass metabolism (or intestinal availability), and the hepatic first-pass metabolism (or hepatic availability). However, for drugs that undergo phase-2 conjugate metabolism and enterohepatic circulation, metabolism by intestinal bacteria is also involved” that too without proper citation to literature”

The corresponding sentences were changed by citing 2 references newly (Page 2).

For enterohepatic circulation, some comments about efflux transport, MRPs, were added (Page 10)

  • Supplement heading 2 with more studies regarding interaction of exo- and endogenous substances with bacteria inhabiting GI tract.

A short comment was added, because, conjugated metabolites such as dietary polyphenols, NSAIDs, and bile acids are representative exo- and endogenous substances that are subject to enterohepatic circulation and interact with intestinal bacteria (Page 2).

  • Rephrase and rewrite the starting information under heading 3 i.e., In clinical, to diagnose SIBO with an increased number and/or altered type of bacteria in the upper gastrointestinal tract, the quantitative culture of duodenum/jejunum fluid aspirates is conducted.

The corresponding sentences were eliminated.

  • Take proper care of citation in the text e.g., [12.34]

This was corrected to [12, 34]

  • Supplement sub-heading 3.1 with new information regarding cultural analysis at start of the sentence. Also, include some aspects of collecting the duodenum/jejunum aspirate.

Information on plasma bacterial 16S rRNA/rDNA gene fragments was added in 3. And the sub-deading in 3. and 3.1 were changed.

  • Additionally, rephrase the statement “Increases in Gammaproteobacterial………” in sub-heading 3.2

The sentence was corrected

  • In 3.3 sub0heading information is pertaining to 16SrRNA and in 3.4, there is sentence regarding 16SrDNA…. Please clarify

 Both 16SRNA/DNA are used for identification of type and number of bacteria.

  • In sub-heading 3.3, please repharse the sentence and try to write in 2 sentences so that the meaning gets clarity.

Fecal microflora was changed to fecal bacteria

  • In heading 7, there is information in one para which is too lengthy to get the desired meaning an d information. Please rephrase the sentence and try to write in 2 or 3 sentences so that the meaning gets clarity.

The contents in heading 7, were divided into 3 sub-headings.

  • The same problems persists in other sections as well. It is advised to revise the manuscript thoroughly and supplement different sections of the paper with updated information and proper reference to literature.

Sentences and contents were revised greatly. They are written in red.

  • Section conclusion: The section needs to be a bit more elaborative and should highlights importance of the study and future directions with possible limitations.

My other suggestion is supplement the paper with some more tables and appropriate figures.

Discussion section and conclusion section was revised greatly in red.

Also, tables and figures were corrected.

Reviewer 4 Report

The work by Maeda and Murakami on "Diagnosis of Small Intestinal Bacterial Overgrowth using Bile Acid Conjugates" is an extremely interesting piece of literature that requires a few editorial changes:

The figure 1 caption should be below the figure and not above it. In addition, I am asking you to increase its quality.

I would like to ask you to check the text for the spelling of bacterial names in italics, because I noticed the lack of it in several places, but due to the lack of line numbers in the manuscript, I am unable to cite specific examples here.

The figure 2 caption should be below the figure itself. In addition, I would like to ask you to explain the abbreviations used in the figure in its description, it will allow for its better reception.

Please explain why table 2 is on the last page? Moreover, it is not quoted anywhere in the main text of the manuscript.

The work is extremely interesting, however, it seems to me that as a review, part of the text could be replaced with pictures, which significantly diversified the work and its reception. In addition, a short fragment explaining what SIBO is and what is its epidemiology and causes would be useful, which would create a logically coherent text with the part describing diagnostics.

Additionally, I would like to extend the sentence "Literature was searched using PubMed and appropriate keywords such as SIBO and diagnosis." What were the criteria for searching, inclusion and exclusion of papers for review? Was only one database used? For this purpose, it may be useful to perform a PRISMA analysis.

Author Response

To Reviewer 4

Thank you very much for your suggestion/indication. We corrected our manuscript according to your suggestion/ indication. Please check the red characters in the revised manuscript.

The title of the manuscript was changed to “Diagnosis by microbial culture, breath tests and urinary excretion tests, and treatments of small intestinal bacterial overgrowth” (Page 1st) Corrections were made in the revised manuscript in red.

  • The figure 1 caption should be below the figure and not above it. In addition, I am asking you to increase its quality.

The correction was made for the caption and figure for Fig. 1

  • I would like to ask you to check the text for the spelling of bacterial names in italics, because I noticed the lack of it in several places, but due to the lack of line numbers in the manuscript, I am unable to cite specific examples here.

 The bacterial name was changed to italics throughout the text.

  • The figure 2 caption should be below the figure itself. In addition, I would like to ask you to explain the abbreviations used in the figure in its description, it will allow for its better reception.

The correction was made for Fig. 2.

  • Please explain why table 2 is on the last page? Moreover, it is not quoted anywhere in the main text of the manuscript.

The position of Table 2 was changed (Page 17)..

  • The work is extremely interesting, however, it seems to me that as a review, part of the text could be replaced with pictures, which significantly diversified the work and its reception. In addition, a short fragment explaining what SIBO is and what is its epidemiology and causes would be useful, which would create a logically coherent text with the part describing diagnostics.

In this article, we analyzed the fate of SIBO diagnostic substrates, especially of bile acid-conjugates, pharmacokinetically, by considering the modulation of small intestinal bacteria (overgrowth) as the influencing factor on pharmacokinetics. We are not medical doctors, and our expertises are in Biopharmaceutics (Murakami T) and Drug Design (Maeda Y), respectively, Pharmaceutical Sciences. We prepared this manuscript from the viewpoint of the pharmacokinetics (PK) of diagnostic substrates and the modulation of PK by intestinal bacteria (SIBO) as an influencing factor. Abstract and introduction sections were revised to make clear the aim of our article from the viewpoint of pharmacokinetics (Page 2nd).

  • Additionally, I would like to extend the sentence "Literature was searched using PubMed and appropriate keywords such as SIBO and diagnosis." What were the criteria for searching, inclusion and exclusion of papers for review? Was only one database used? For this purpose, it may be useful to perform a PRISMA analysis.

The corresponding sentence was altered to “Literature was searched using PubMed, Google Scholar, Online search engines and appropriate keywords such as SIBO and diagnosis”.(Page 2).

Round 2

Reviewer 1 Report

Thanks to the authors for answering the questions. I have no other questions.

Author Response

Thank you very much for your kind review of our revised manuscript.

We revised the Discussion Section again.

Reviewer 2 Report

Thanks for considering the suggestions, no further comments. The discussion can be further optimized in terms of the key message as well as the organization of some of the presented information.

Author Response

To Reviewer 2

Thank you very much for your kind comments to improve  our manuscript.

We revised the Discussion section extensively according to your suggestion.

The revision was made in red highlighted with yellow  on manuscript.

Please check the revision.

Thank you very much again for reviewing our manuscript.   
